# Genome-Wide Association Study on Body Conformation Traits in Xinjiang Brown Cattle

**DOI:** 10.3390/ijms251910557

**Published:** 2024-09-30

**Authors:** Menghua Zhang, Yachun Wang, Qiuming Chen, Dan Wang, Xiaoxue Zhang, Xixia Huang, Lei Xu

**Affiliations:** 1College of Animal Science, Xinjiang Agricultural University, Urumqi 830052, China; zhangmenghua810@126.com (M.Z.); cqm19860612@126.com (Q.C.); wangdan01100330@163.com (D.W.); zhangxiaoxue0726@163.com (X.Z.); 2Laboratory of Animal Genetics, Breeding and Reproduction, Ministry of Agriculture of China, National Engineering Laboratory of Animal Breeding, College of Animal Science and Technology, China Agricultural University, Beijing 100193, China; wangyachun@cau.edu.cn

**Keywords:** Xinjiang Brown cattle, body conformation traits, genome-wide association study, quantitative trait loci

## Abstract

Body conformation traits are linked to the health, longevity, reproductivity, and production performance of cattle. These traits are also crucial for herd selection and developing new breeds. This study utilized pedigree information and phenotypic (1185 records) and genomic (The resequencing of 496 Xinjiang Brown cattle generated approximately 74.9 billion reads.) data of Xinjiang Brown cattle to estimate the genetic parameters, perform factor analysis, and conduct a genome-wide association study (GWAS) for these traits. Our results indicated that most traits exhibit moderate to high heritability. The principal factors, which explained 59.12% of the total variance, effectively represented body frame, muscularity, rump, feet and legs, and mammary system traits. Their heritability estimates range from 0.17 to 0.73, with genetic correlations ranging from −0.53 to 0.33. The GWAS identified 102 significant SNPs associated with 12 body conformation traits. A few of the SNPs were located near previously reported genes and quantitative trait loci (QTLs), while others were novel. The key candidate genes such as *LCORL*, *NCAPG*, and *FAM184B* were annotated within 500 Kb upstream and downstream of the significant SNPs. Therefore, factor analysis can be used to simplify multidimensional conformation traits into new variables, thus reducing the computational burden. The identified candidate genes from GWAS can be incorporated into the genomic selection of Xinjiang Brown cattle, enhancing the reliability of breeding programs.

## 1. Introduction

The breeding industry of Xinjiang Brown cattle accounts for a large proportion of the local economic development as well as farmers’ and herders’ income. In 2023, the population of Xinjiang Brown cattle reached 2 million. The development of Xinjiang Brown cattle, a dual-purpose breed, is challenged by the limited scale of linear type traits evaluation, incomplete genetic evaluation systems, and underutilization of genomic breeding technologies. The Xinjiang conformation evaluation method integrates linear scoring from dairy breeds and specific conformation traits from beef breeds. Most countries employ a 9-point scoring system for conformation traits, with the German Brown cattle system encompassing 25 traits across five categories: body frame, muscularity, rump, feet and legs, and mammary system. The latest Chinese dual-purpose cattle total performance index allocates 10% to conformation traits, highlighting their importance in breeding. Conformation traits, while non-productive, are correlated with economically significant traits, such as milk yield [1], reproductive performance [2], health [3], economic efficiency [4], and longevity [5]. Therefore, selecting for body size traits can accelerate genetic progress in other economic traits. Body size appearance identification is also indispensable for selection and breeding [6], where assessors analyze identification data to determine herd defects and formulate selection and mating plans [7,8].

Genetic parameters of body size traits are important for formulating and implementing breeding plans. There are phenotypic and genetic correlations among body conformation traits [9]. For example, Mazza et al. found strong correlations between rear udder height and width [10], while Wang Dan et al. detected a significant correlation between the body structure traits of Xinjiang Brown cattle [11]. The correlations between body size traits indicate redundant information among traits, and reducing the analysis of highly correlated traits is a concern for breeders because, in breeding selection, the highly correlated traits can be selected indirectly from the remaining highly correlated traits by selecting one of them. Principal component analysis (PCA) and factor analysis can explore relationships among traits by reducing data dimensions with minimal information loss [12]. These methods can transform the linear combination of original variables into composite variables using variable weights obtained from the correlation matrix of the original data [13].

Genetic analysis of complex traits, such as body conformation in cattle, is a challenging problem in animal genetic breeding. Due to its recent advances and affordability, resequencing technology has become essential for identifying the loci associated with complex traits in cattle. A genome-wide association study (GWAS), based on linkage disequilibrium, identifies molecular markers associated with phenotypic variations in complex traits by screening thousands of high-density molecular markers within a population. GWAS has been successfully applied in animal breeding [14,15,16,17,18]. Zhou Jinghang et al. reported 12 SNPs associated with milk production and reproductive performance in Xinjiang Brown cattle in 2019 [19]. Globally, the selection indices in countries such as the USA, Canada, Australia, and Europe include body conformation traits as breeding targets [20]. The genetic evaluation models can be continuously optimized by adding SNPs associated with breeding target traits identified through GWAS. Currently, genomic genetic evaluation for dual-purpose cattle in China is still underdeveloped, and molecular markers related to body size traits need to be investigated.

The evaluation method of body conformation traits of Xinjiang Brown Cattle (dual-purpose breed) is formed by absorbing the linear scoring method of Chinese Holstein cattle (dairy breed), referring to the “code of practice of type classification in Chinese Holstein”, and introducing the special body size evaluation method of meat breeds. However, these details have been scarcely researched. Given the need to explain different biological meanings and avoid redundant information among body conformation traits, this study aims to estimate and compare the genetic parameters of Xinjiang Brown cattle using factor analysis. A GWAS was also conducted on body conformation traits based on whole-genome resequencing data. This research provides a scientific basis for the selection and breeding of Xinjiang Brown cattle.

## 2. Results

### 2.1. Descriptive Statistics, Variance Components, and Heritability Estimates of Body Size Traits

Table 1 lists the descriptive statistics, variance components, and heritability estimates of body conformation traits in Xinjiang Brown cattle. The top three body conformation traits with the highest coefficients of variation were rump angle, udder depth, and central ligament, indicating large individual differences in their phenotypic values. The heritability estimates of the body conformation traits ranged from 0.01 to 0.59. Front teat diameter and udder balance showed low heritability, while other body conformation traits exhibited medium to high heritability.

### 2.2. Factor Analysis Result

Table 2 lists the eigenvalues and proportions of phenotypic variance explained by each factor in the Xinjiang Brown cattle. Eight latent factors with eigenvalues > 1 explained 59.12% of the information, with the first factor (F1) accounting for the largest proportion of total variance at 16.14%. Figure 1 shows the factor loadings obtained using the varimax rotation method. In F1, rump height (0.69), body depth (0.67), and rump width (0.67) had high loadings, reflecting the body frame and rump information. In F2, foot angle (0.64) and heel depth (0.63) had high loadings, reflecting the limb and hoof information. F3, indicating udder length information, had high loadings for the lengths of the front (0.71) and rear (0.76) udders. F4, reflecting teat position information, had high loadings for the front (0.82) and rear (0.81) teat positions. F5 mainly reflected muscle development information, with loadings of 0.62 and 0.85 for rear leg girth and withers width, respectively. F6, mainly reflecting teat size information, had high loadings for front teat diameter (0.76) and front teat length (0.81). F7 and F8 mainly reflected udder traits.

### 2.3. Heritability and Genetic Correlation of Factor Scores

Table 3 shows the variance component and heritability estimate for all eight factors in Xinjiang Brown cattle, with the latter ranging from 0.17 (F6) to 0.73 (F2), all being medium to high heritability traits ($h^{2}$ > 0.1).

### 2.4. Rank Correlation between Estimated Breeding Values of Body Conformation Traits and Factor Scores

Figure 1 shows the rank correlation coefficients between the estimated breeding values (EBVs) of body conformation traits and factor scores in Xinjiang Brown cattle. Notably, the rank correlation coefficients have similar patterns to the factor loadings of body conformation traits in Figure 2. The results indicate that the EBVs of F1 had high positive correlations with the EBVs of two body frame traits and two rump traits. The EBVs of F2 in Xinjiang Brown cattle had high correlations with those of two feet and legs traits, showing similar research results to Figure 2. Other factors in this study showed similar results; for example, the correlations for the EBVs of F3 were similar to the loading coefficients of body conformation traits in Figure 2. Similar results were observed in F3, F4, F5, F6, F7, and F8.

Figure 3 shows the first three principal components, accounting for 11.89%, 8.23%, and 5.93% of the total variance, respectively. By comparing the first three principal components (PCs), individuals from farm 1 (black) and farm 2 (red) could be distinguished. Although the 496 individuals came from different farms, some overlaps existed among farms, such as between farm 3 (blue) and farms 1 (black) and 2 (red), and between farm 3 (green) and farms 5 (purple) and 6 (yellow). The PCA results of 496 lactating Xinjiang Brown cattle showed some genetic connections and distances among the farms.

### 2.5. Population Analysis of Body Conformation Traits

Figure 4 shows that the GWAS of body conformation traits based on the single-trait mixed linear model in GEMMA v0.98.5 software had no systemic bias. The model considered population effects and kinship, so the PCs were not included in the model. Additionally, the genomic inflation factors of body conformation traits in Xinjiang Brown cattle ranged from 0.859 to 1.022, indicating no significant genomic inflation.

### 2.6. QTL Mapping of Body Conformation Traits in Xinjiang Brown Cattle

#### 2.6.1. QTL Mapping of Body Conformation Traits

GWAS analysis of body conformation traits (stature, body depth, chest width) in Xinjiang Brown cattle identified 142, 0, and 2 significant SNPs. SNPs significantly associated with rump height and chest width were located on chromosomes 3 and 6 (Figure 5) and chromosome 6 (Figure 6), respectively.

The SNP with the lowest *p*-value within each locus was defined as the lead (Table 4). According to the annotation by ANNOVAR software, the most significant locus for rump height was located on chromosome 6, with six leading SNPs in the intronic regions of the *LAP3*, *MED28*, *LCORL*, and *NCAPG* genes and one leading SNP upstream and downstream of *IBSP* and *TRNAA-CGC*, explaining 9%–11.3% of the phenotypic variance. The most significant locus for chest width was on chromosome 3, with the leading SNP in the intronic region of *NOS1AP*, explaining 9.1% of the phenotypic variance.

#### 2.6.2. QTL Mapping of Muscle Development Traits

GWAS analysis of muscle development traits in Xinjiang Brown cattle identified 13, 21, 707, and 0 significant SNPs. The significantly associated SNPs were located on chromosome 1 (Figure 6) for withers width and chromosomes 3, 5, 8, 11, 12, 13, 14, 18, 19, and 21 (Figure 6) for rear leg girth. The SNPs for rear leg height were on all chromosomes except 27 and 28, with the most significant SNPs (118) on chromosome 1 and strong signals observed on chromosome 13 (Figure 6).

The SNP with the lowest *p*-value within each locus was defined as the leading SNP (Table 5). According to the annotation by ANNOVAR software, the most significant locus for withers width was located on chromosome 1, with leading SNPs upstream and downstream of *DPPA4* and *TRNAE-UUC*, explaining 11.6% of the phenotypic variance. The most significant locus for rear leg girth was located on chromosome 13, with two leading SNPs upstream and downstream of *CDH4*, *CDH26*, *LOC112449365*, and *BTBD3*, explaining 8.6% and 7.6% of the phenotypic variance, respectively. The 51 leading SNPs significantly associated with rear leg height explained 6.9% to 17.1% of the phenotypic variance, with the most significant SNPs upstream and downstream of *SYT13* and *LOC112441655*.

#### 2.6.3. QTL Mapping of Rump Traits

GWAS analysis of rump traits in Xinjiang Brown cattle identified 216, 81, and 0 significant SNPs. The SNPs significantly associated with rump length were distributed on all chromosomes except 18, 19, 25, 28, and 29, with the most significant SNPs on chromosome 24 (98) and strong signals on chromosome 24 (Figure 7). SNPs significantly associated with rump width were distributed on all chromosomes except 10, 13, 16, 17, 18, 20, 21, 25, and 29, with the most significant SNPs on chromosome 7 (17) (Figure 7).

The SNP with the lowest *p*-value within each locus was considered the lead (Table 6). According to the annotation by ANNOVAR software, the most significant locus for rump length was located on chromosome 24, with three leading SNPs upstream and downstream of *LOC112444152*, *TRNAC-GCA*, *TRNAK-UUU*, and *CDH7*, explaining 10.6% to 14.7% of the phenotypic variance. The most significant locus for rump width was located on chromosome 23, with the leading SNP in the intronic region of *OFCC1*, explaining 12.6% of the phenotypic variance.

#### 2.6.4. QTL Mapping of Limb Traits

GWAS analysis of limb traits in Xinjiang Brown cattle identified 0, 0, 21, and 11 significant SNPs (*p* < 3.18 × 10^−9^). SNPs significantly associated with hock texture were distributed on chromosomes 8, 11, 18, 22, and 26, with the most significant SNPs on chromosome 26 (11) (Figure 8). SNPs significantly associated with the rear view of the rear legs were distributed on chromosomes 2 and 3, with the most significant SNPs on chromosome 3 (10) (Figure 9).

The SNP with the lowest *p*-value within each locus was considered the leading SNP (Table 7). According to the annotation by ANNOVAR software, the most significant locus for hock texture was located on chromosome 8, with the leading SNP in the intronic region of *ABCA1*, explaining 9.7% of the phenotypic variance. Two leading SNPs on chromosome 26 were upstream and downstream of *ADRA2A, GPAM*, *MKI67*, and *MGMT*, explaining 8.8% and 7.7% of the phenotypic variance, respectively. Another leading SNP on chromosome 26 was in the intronic region of *KNDC1*, explaining 7% of the phenotypic variance. For the rear view of the rear legs, the leading SNP on chromosome 3 was upstream and downstream of *KCNN3* and *ADAR*.

#### 2.6.5. QTL Mapping of Udder Traits

GWAS analysis of udder traits in Xinjiang Brown cattle identified 1, 325, and 4 significant SNPs for rear udder length, front teat length, and rear teat position, respectively. The significantly associated SNPs were located on chromosome 24 for rear udder length and on all chromosomes except 3, 12, and 19, with the most significant SNPs on chromosome 7 for front teat length (Figure 9). SNPs significantly associated with rear teat position were distributed on chromosomes 1 and 7, with three significant SNPs on chromosome 1 and one on chromosome 7.

The SNP with the lowest *p*-value within each locus was defined as the lead (Table 8). According to the annotation by ANNOVAR software, the most significant locus for rear udder length was located on chromosome 24, with the leading SNP upstream of *DLGAP1*, explaining 7.1% of the phenotypic variance. For front teat length, two leading SNPs with strong signals on chromosome 7 were upstream and downstream of *RGMB* and *CHD1*, explaining 12.0% and 11.4% of the phenotypic variance, respectively. Another leading SNP was in the intronic region of *FBXL17*. For the rear teat position trait, two leading SNPs on chromosomes 7 and 1 were upstream and downstream of *ELL2*, *PCSK1*, and *CADM2*.

## 3. Materials and Methods

### 3.1. Data Source and Processing

The body conformation data for Xinjiang Brown cattle were collected from nine core breeding farms in Xinjiang from July 2018 to July 2022. A total of 27 body size traits were measured, including 17 measurable traits and 10 scored traits (1–9 points). The specific assessment parts are shown in Figure 10. Scored traits were assessed by two certified Chinese dairy cattle body size assessors (certificate numbers: DAC-TY-083, DAC-TY-019). A total of 1816 lactating Xinjiang Brown cows were assessed, with abnormal records excluded based on the principle of “mean ± 3 standard deviations” and cows without pedigree records removed. Finally, the data from 1185 lactating Xinjiang Brown cows were further analyzed. The pedigree records of Xinjiang Brown cattle consisted of 186 bulls and 2067 cows, checked and sorted using the chkPed() function of ASReml v4.2 software (VSNC, Asian-Pacific region, UK).

### 3.2. Blood Sample Collection and Whole-Genome Resequencing

Blood samples (10 mL) were collected from the tail vein of the test animals and the anticoagulated whole blood was stored at −20 °C for genomic testing. Genomic DNA was extracted from 496 Xinjiang Brown cattle and sequenced using the MGISEQ-2000 platform by BGI (MGI Tech, Shenzhen, China). The resequencing of 496 Xinjiang Brown cattle generated approximately 74.9 billion reads, with an average sequencing depth of 8.04× and an average alignment rate of 99.87%.

### 3.3. Read Alignment and SNP Detection

The raw reads from the resequencing data of 496 Xinjiang Brown cattle were first quality-controlled using Trimmomatic v0.40 software (http://USADELLAB.org) [21]. Clean reads were aligned to the ARS-UCD1.2 reference genome using the MEME module of BWA v0.7.17 software [22] with default parameters. Duplicate reads were removed using the Sortsam and MarkDuplicates modules of Picard v2.25.5 software to reduce PCR amplification bias. Base quality score recalibration was performed using the BaseRecalibrator and PrintReads modules of GATK. The SNPs were detected using the HaplotypeCaller, CombineGVCFs, GenotypeGVCFs, and SelectVariants modules of GATK v3.8 [23]. Low-quality SNPs were filtered using the VariantFiltration module with the following parameters: QD < 2.0, FS > 60.0, MQ < 40.0, MQRankSum < −12.5, ReadPosRankSum < −8.0, and SOR > 3.0. SNPs were aligned to the ARS-UCD1.2 reference genome, and functional annotation of filtered SNPs was performed using ANNOVAR software [24]. SNPs were excluded based on the following criteria: (1) individual genotype call rate < 90%, (2) single SNP genotype call rate < 90%, (3) significant deviation from Hardy–Weinberg equilibrium (*p* < 10^−6^), (4) minor allele frequency (MAF) < 0.05, and (5) unknown chromosome or genome information. A total of 15,700,670 SNPs were retained in Xinjiang Brown cattle.

### 3.4. Statistical Analysis

#### 3.4.1. Estimation of Genetic Parameters

Using ASReml v4.2 software, the variance components of 27 body conformation traits, the PCs, and factors were estimated using the average information restricted maximum likelihood method in an animal model. The model used was:Y=Xβ+Za+e
where Y represents the observed vector of 27 body conformation traits, *β* is the fixed effects vector including farm effect (9 levels), days in milk effect (9 levels: 10–40 days, 41–80 days, 81–120 days, 121–160 days, 161–200 days, 201–240 days, 241–280 days, 281–320 days, >320 days), and parity effect (4 levels: 1st parity, 2nd parity, 3rd parity, ≥4th parity), a is the additive genetic effect vector, e is the random residual, and X and Z are the incidence matrices for fixed and random effects, respectively.

A two-trait animal model was constructed using PCs and factor scores as cap *Y* to estimate their genetic correlations. The two-trait animal model matrix form is:$$\left[\begin{array}{c} y_{1}\\ y_{2} \end{array}\right]=\left[\begin{array}{cc} X_{1} & 0\\ 0 & X_{2} \end{array}\right]\left[\begin{array}{c} b_{1}\\ b_{2} \end{array}\right]+\left[\begin{array}{cc} Z_{1} & 0\\ 0 & Z_{2} \end{array}\right]\left[\begin{array}{c} g_{1}\\ g_{2} \end{array}\right]+\left[\begin{array}{c} e_{1}\\ e_{2} \end{array}\right]$$
where *y*_1_ and *y*_2_ are the observed values of each factor and principal component, *b*_1_ and *b*_2_ are the fixed effects vectors, with fixed effects consistent, *g*_1_ and *g*_2_ are the additive genetic effects vectors, *e*_1_ and *e*_2_ are the random residuals, and *X*_1_, *X*_2_, *Z*_1_, and *Z*_2_ are the corresponding incidence matrices.

The estimated variance components were used to calculate the genetic parameters [25]:$$\mathrm{Heritability}:\;h^{2}=\frac{\sigma_{a}^{2}}{\sigma_{a}^{2}+\sigma_{e}^{2}};\;\mathrm{Genetic}\;\mathrm{correlation}:\;r_{A}=\frac{Cov\left(a_{i},a_{j}\right)}{\sqrt{\sigma_{i}^{2}\sigma_{j}^{2}}}$$
where $h^{2}$ is heritability, $\sigma_{a}^{2}$ is additive genetic variance, $\sigma_{e}^{2}$ is residual variance, and $Cov\left(a_{i},a_{j}\right)$ is the additive genetic covariance between two traits.

#### 3.4.2. Factor Analysis

Factor analysis was conducted using the FACTOR procedure in SAS v9.1 software, which synthesizes the information contained in a set of *n* observed variables by seeking a set of new variables, known as common latent factors. The varimax rotation method was used to maintain the orthogonality of extracted factors, retaining only factors with eigenvalues ≥1 [26]. Factor loadings of each body conformation trait were observed to interpret analysis results biologically. Standardized factor scores were calculated for each cow to facilitate the analysis of common factors as analysis variables instead of the original variables before dimension reduction.

As described previously [27], the factor analysis model is expressed as:$$y_{n}=b_{n1}X_{1}+b_{n2}X_{2}+\cdots b_{np}X_{p}+e_{n}$$
where *y_n_* is the *n*th original variable, *b_np_* is the loading of each variable *n* on each factor, *x_p_* is the *p*th common factor of the nth variable, and *e_n_* reflects the specific factor of the *n*th variable.

#### 3.4.3. Genome-Wide Association Study

Haplotype inference and missing allele imputation for GWAS analysis were performed using the method described in BEAGLE v4.1 software [28], which included a total of 15,700,670 SNPs for Xinjiang Brown cattle.

Here, GWAS analysis for 27 body conformation traits of Xinjiang Brown cattle was conducted using the single-trait mixed linear model in GEMMA v0.98.5 software [29]. The GWAS model is:y=Wα+Xβ+Kμ+ε
where Y represents the observed vector of 27 body conformation traits, W is the covariant matrix, including farm effect (9 levels), days in milk effect (9 levels: 10–40 days, 41–80 days, 81–120 days, 121–160 days, 161–200 days, 201–240 days, 241–280 days, 281–320 days, >320 days), and parity effect (4 levels: 1st parity, 2nd parity, 3rd parity, ≥4th parity), α is the covariant vector, X is the marker genotype, β is the marker genotype effect; K is the genetic relationship coefficient matrix, μ is the genetic relationship coefficient effect, and ε is the random residual vector.

Bonferroni correction was applied to multiple hypothesis testing, setting the significance threshold *p*-value at 3.18 × 10^−9^. GWAS visualization was performed using the CMplot package in R v4.4.1 software.

To further determine the genetic effects of significant SNPs, the proportion of phenotypic variance explained by significant SNPs was calculated using the formula reported by Shim et al. [30]:$$PVE=\frac{2{\hat{\beta }}^{2}MAF(1-MAF)}{2{\hat{\beta }}^{2}MAF(1-MAF)+{\left[se(\hat{\beta })\right]}^{2}2NMAF(1-MAF)}$$
where ${\hat{\beta }}^{2}$ is the effect size of the SNP marker, MAF is the minor allele frequency, $se\left(\hat{\beta }\right)$ is the residual variance, and *N* is the sample size for GWAS analysis.

Genomic inflation factor λ was calculated using R v4.4.1 software, converting association test statistics *p*-values to $\chi^{2}$, and dividing the median $\chi^{2}$ statistic by the expected median of the $\chi_{1}^{2}$ distribution (0.4549).

After completing the GWAS analysis, linkage disequilibrium among significant SNPs was calculated, and the critical value for associated loci was defined. The LD is denoted by *R*^2^ ($R^{2}=\frac{D^{2}}{P\left(A_{1}\right)P\left(A_{2}\right)P\left(B_{1}\right)P\left(B_{2}\right)},\;D=P\left(A_{1}B_{1}\right)-P\left(A_{1}\right)P\left(B_{1}\right)$). Loci with fewer than three significant SNPs were excluded from further analysis to reduce false-positive signals. The SNP with the smallest *p*-value within each locus was defined as the leading SNP. Functional annotation of SNPs related to body size traits was performed using ANNOVAR software [24] based on the ARS-UCD1.2 reference genome. Candidate genes within a 500 Kb range upstream and downstream of significant SNPs were considered for further analysis [30].

## 4. Discussion

### 4.1. Estimation of Genetic Parameters for Body Size Traits and Factor Scores

The rump height of lactating Xinjiang Brown cattle was similar to that of Swiss Brown cattle (139 cm) [31], whereas the body depth was similar to that of Slovenian Brown cattle (76.20 cm) [32]. The rump length and width were higher than those of Slovenian Brown cattle (48.40 cm and 17.59 cm, respectively) [32]. These two body conformation traits had medium to high heritability, but chest width had low heritability. Several studies have reported lower heritability for chest width compared with rump height and body depth [33]. Rump traits showed higher heritability, consistent with studies on Italian Brown cattle [34], while most feet and legs traits have been reported to have low to medium heritability [33,35]. This study analyzed more body conformation traits than previous studies in different cattle breeds [36,37], somewhat increasing the data dimension. Generally, cows with deep and tall bodies and wide and large udders are better milk producers [31,38]. Body conformation traits with high loadings in F1 are usually associated with milk production performance, consistent with the findings of Kern et al. [38]. Thus, F1 can be included in selection indices to improve milk production performance.

In this study, the rank correlation results between the EBVs of body conformation traits and the factor scores exhibited patterns similar to those of the factor loadings of body conformation traits. Mazza S et al. reported similar results in two dual-purpose cattle breeds [39]. This further indicates the consistency of using factor score phenotypes for genetic evaluation of body conformation parts. Although factor analysis can eliminate redundant information in correlated variables, it is ambiguous in explaining single traits. When conducting genetic selection for multiple traits in a population, the factor score can effectively eliminate redundancy among traits of interest under genetic selection.

### 4.2. QTL for Body Frame Traits

Body frame traits include stature, body depth, and chest width. In this study, *LCORL* and *NCAPG* were significantly associated with stature in Xinjiang Brown cattle. An Bingsing et al. reported a significant association between *LAP3*, which regulates hormone secretion levels and protein maturation [40], and stature in Wagyu cattle [41]. Additionally, *LAP3* and *FAN184B* have been reported to be significantly associated with bone weight in beef cattle [42,43]. Shen Jiafei et al. found that *IBSP* is significantly associated with the ear margin area in Brahman and Yunling cattle [44]. *LCORL* encodes a transcription factor that may function in spermatogenesis in the testes [45], while *NCAPG* encodes a non-histone chromatin-associated protein G in mammals, which is part of the condensin I complex involved in chromatin compression and regulation, especially during mitosis [46]. *LCORL* and *NCAPG* are key genes affecting height in many species, such as horses [18], cattle [47], and dogs [48]. *LCORL* and *NCAPG* have been identified as height-related loci in European, Japanese, and African populations [49,50,51,52]. However, due to the proximity of *LCORL* and *NCAPG* and high linkage disequilibrium among SNPs in these regions, it is still uncertain which gene has a greater impact on height, requiring further verification.

### 4.3. QTL for Muscle Development Traits

Candidate genes related to muscle development traits include *CPNE4*, *RUS1*, *DNAJC1TBX18*, *REG3G*, *AHCYL2*, *FSTL5*, *NPFFR2*, and *FAT3*, which have been reported to be associated with growth, muscle development, body size, and meat performance. *CPNE4* is a gene related to glycogen content, regulating muscle glycogen content by affecting glucose metabolism and being associated with growth performance, body size, muscle, and skeletal development in cattle [53,54,55]. *RUS1* was shown to influence weaning weight in sheep [56]. *DNAJC1* was reported to have pleiotropy in weaning and yearling weight in Angus beef cattle [57]. However, *TBX18* has not been previously found to be related to body size traits. However, a recent study showed an association between *TBX18* and growth performance in Simmental dual-purpose cattle [58]. Additionally, *TBX18* is related to obesity in humans and mice [59,60]. Lee et al. reported that *TBX18* regulates processes related to skeletal muscle metabolism, affecting body size in animals [61]. Studies on Nellore cattle showed that *REG3G* is associated with growth performance [62], while *AHCYL2* is related to carcass backfat thickness [63]. In a study on Limousin and Jersey cattle crossbreeds, Novianti et al. reported that *FSTL5* on bovine chromosome 17 is a key gene affecting muscle development [64]. While *NPFFR2* was shown to be significantly associated with the udder morphology traits of German Simmental cattle [65], it was significantly associated with withers width in this study. Regarding meat performance, Riley et al. found that the *FAT3* gene is related to beef palatability [66].

### 4.4. QTL for Rump Traits

*EBF1*, *NSMCE2*, *TASOR*, *NEDD4*, and *PGF*, associated with rump traits in Xinjiang Brown cattle, are related to cattle reproduction and rump traits. Cole et al. reported that *EBF1* affects rump width and calving ease in Holstein cattle [67]. Rump width has been used as an indirect selection trait for calving ease. The bovine QTL database search results also showed that *EBF1* is associated with rump width and calving ease. In this study, this gene was associated with rump traits in Xinjiang Brown cattle, making it a candidate gene for rump trait selection. Moreover, in addition to affecting rump width in dairy cattle [67], *NEDD4* is also related to the eye muscle area in beef cattle. Cao et al. demonstrated that *NEDD4* deficiency reduces IGF-1 and insulin signaling in knockout mice [68]. *PGF*, a member of the vascular endothelial growth factor family, is related to early embryo development in yaks [69]. In a study on German Holstein cattle, *PGF* was shown to be associated with calving ease and stillbirth [70].

### 4.5. QTL for Feet and Legs Traits

In the GWAS study of feet and legs traits in Xinjiang Brown cattle, six candidate genes were annotated, with *MGMT* being noteworthy for its association with body size, reproduction, and longevity traits, including overall body score, rear legs, rear view, limb score, and hoof angle. The loci associated with *MGMT* showed the most significant effect on limb traits [67]. Recent studies have reported that *GATB* is related to white-line disease in dairy cattle [71]. The white line is a soft keratinized area at the junction of the hoof sole and wall. Pérez-Cabal et al. indicated that foot angle significantly influences the incidence of white-line disease [72], with higher linear scores and larger angles being associated with a lower disease incidence. Other candidate genes identified in this study have not been reported in limb-related traits but are considered important candidate genes in cattle behavior and milk production traits. Michenet et al. [73] found that *ADRA2* is associated with maternal protective behavior in cows and encodes an adrenergic receptor related to various behavioral traits in humans, mice, and rats [74]. Hence, this might be a candidate gene affecting cattle behavior.

### 4.6. QTL for Udder Traits

Various GWAS reports on udder traits in different cattle populations have identified several candidate loci and genes [65,75,76,77]. However, studies identifying genomic regions related to udder traits are inconsistent, including this one. These differences in identified genomic regions are possibly influenced by statistical methods and population differences. Although candidate genes annotated to leading SNPs associated with udder traits in this study have been rarely reported, significant SNPs discovered were adjacent to previously reported SNP loci. For example, SNP rs441400818 on *BTA7* and rs380411477 on *BTA17* were close to the SNPs rs29023522 and rs42313276 associated with teat length reported by Cole et al., with annotated genes including *RGMB*, *CHD1*, *CPE*, and *MSMO1*. The SNP rs109736435 on *BTA7* was close to the SNP rs110574421 associated with teat position, annotated to *ELL2* and *PCSK1* [67]. An Animal QTLdb database search indicated that *CHD1*, *CPE*, and *ELL2* are related to a productive life, overall body score, and somatic cell score, directly associated with teat length and position [78,79].

## 5. Conclusions

Most body conformation traits and factor scores in Xinjiang Brown cattle were medium to highly heritable. The rank correlation results between the estimated breeding values of body conformation traits and the factor scores exhibited patterns similar to those of the factor loadings of body conformation traits, suggesting that the factor scores should be included in selection indices. Using latent factors in body conformation trait evaluation can simplify multidimensional body conformation traits into new variables, reducing the computational burden of analyzing large datasets. In Xinjiang Brown cattle, 102 leading SNPs were significantly associated with 12 body conformation traits. Some SNPs were located within or near previously reported genes and QTL regions, whereas others were newly discovered. These significant SNPs identified through GWAS can be included in future genomic genetic evaluation models to continuously optimize these models and provide a scientific basis for the selection and breeding of Xinjiang Brown cattle.

## Figures and Tables

**Figure 1 ijms-25-10557-f001:**
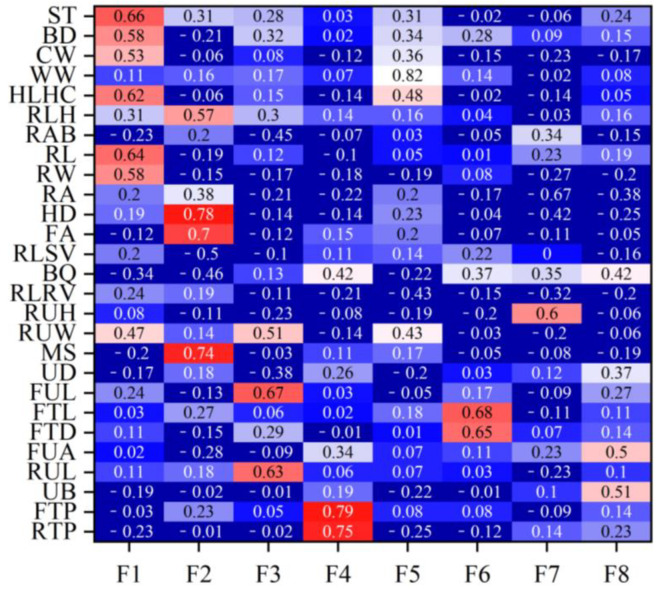
Rank correlation coefficients between the estimated breeding values for each body conformation trait and the estimated breeding values for factor scores for Xinjiang Brown cattle.Note: ST is stature; BD is body depth; CW is chest width; WW is wither width; HLHC is half of leg circumstance; RLH is rear leg height; RAB is rib and bone; RL is rump length; RA is rump angle; HD is heel depth; FA is feet angle; RLSV is rear leg side view; RUH is rear udder height; RUW is rear udder width; MS is medium; UD is udder depth; FUL is fore udder length; FTL is fore teat length; FTD is fore teat diameter; FUA is fore udder attachment; RUL is rear udder length; UB is udder balance; FTP is fore teat placement; RTP is rear teat placement; DC is dairy character; and LS is loin strength. Red means that the correlation coefficient is greater than 0.3, and the darker the color, the greater the correlation coefficient; blue means that the correlation coefficient is less than 0.3, and the darker the color, the smaller the correlation coefficient.

**Figure 2 ijms-25-10557-f002:**
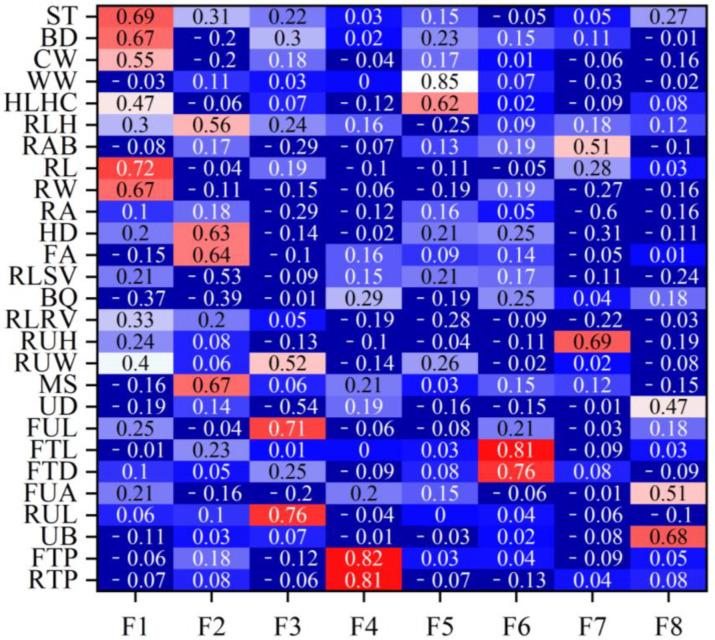
Factor loading coefficients of body conformation traits for Xinjiang Brown cattle. Note: ST is stature; BD is body depth; CW is chest width; WW is wither width; HLHC is half of leg circumstance; RLH is rear leg height; RAB is rib and bone; RL is rump length; RA is rump angle; HD is heel depth; FA is feet angle; RLSV is rear leg side view; RUH is rear udder height; RUW is rear udder width; MS is medium; UD is udder depth; FUL is fore udder length; FTL is fore teat length; FTD is fore teat diameter; FUA is fore udder attachment; RUL is rear udder length; UB is udder balance; FTP is fore teat placement; RTP is rear teat placement; DC is dairy character; and LS is loin strength. Red means that the correlation coefficient is greater than 0.3, and the darker the color, the greater the correlation coefficient; blue means that the correlation coefficient is less than 0.3, and the darker the color, the smaller the correlation coefficient.

**Figure 3 ijms-25-10557-f003:**
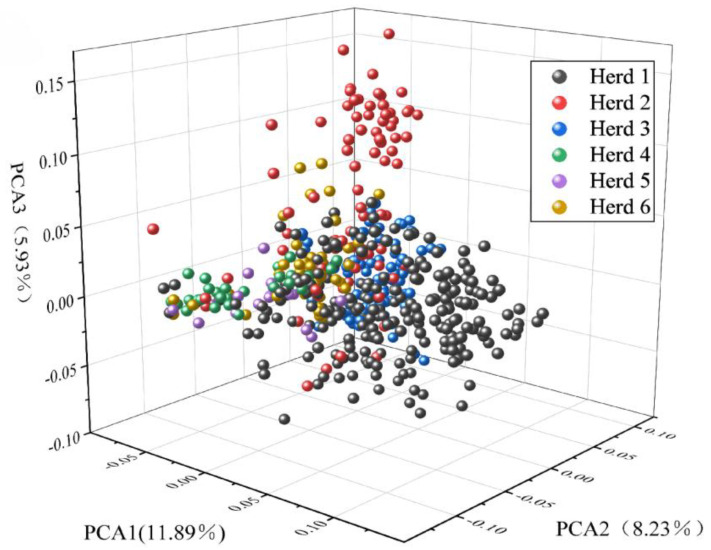
Population structure map showing the first three principal components of Xinjiang Brown cattle.

**Figure 4 ijms-25-10557-f004:**
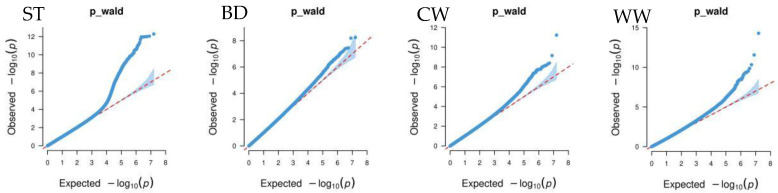
Quantile–quantile (QQ) plot of GWAS analysis for body conformation traits of Xinjiang Brown cattle. Note: ST is stature; CW is chest width; BD is body depth; WW is wither width; HLHC is half of leg circumstance; RLH is rear leg height; RAB is rib and bone; RL is rump length; RW is rump width; RA is rump angle; FA is rump angle; RLSV is rear leg side view; BQ is bone quality; and RLRV is rear leg rear view.

**Figure 5 ijms-25-10557-f005:**
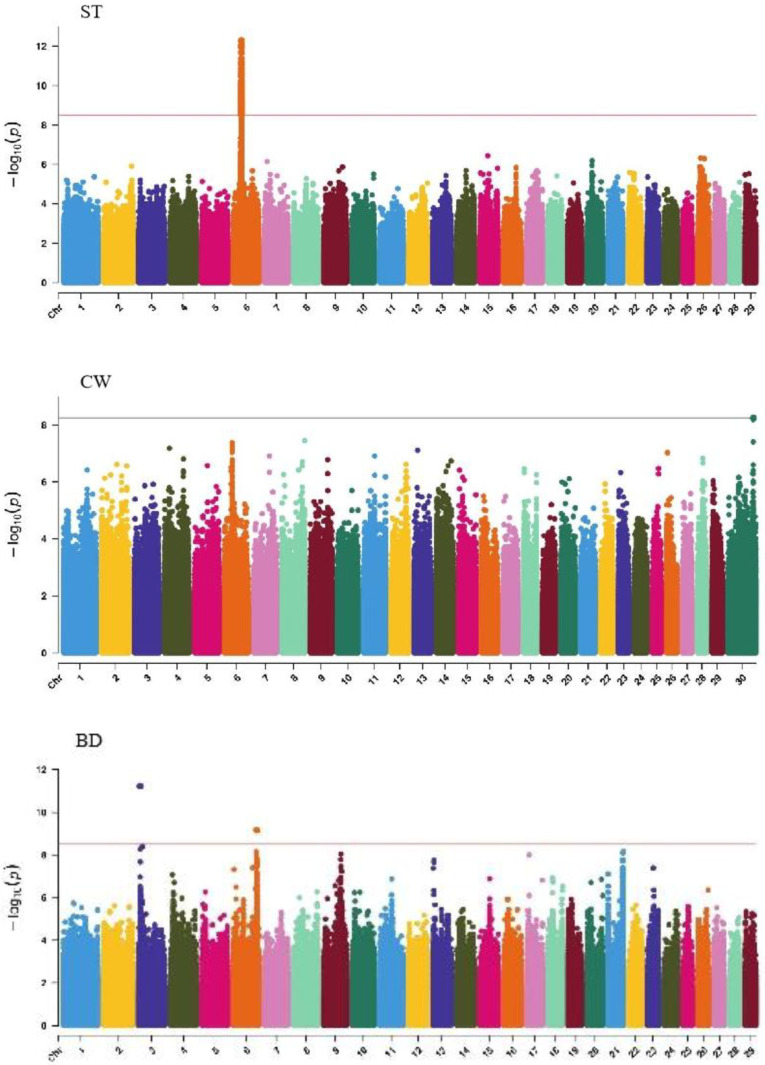
Manhattan plots of body frame traits (stature, body depth, and chest width) in Xinjiang Brown cattle. Note: ST is stature of body frame traits, BD is body depth of body frame traits and CW is chest width body frame traits. Different colors represent different chromosomes, and the number on the horizontal coordinate is the chromosome number.

**Figure 6 ijms-25-10557-f006:**
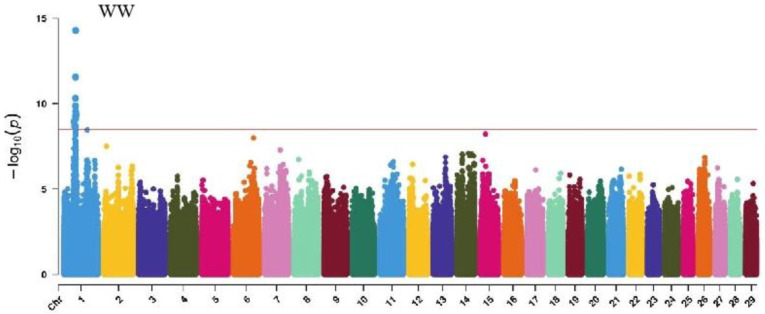
Manhattan plots of muscularity traits (wither width, half of leg circumstance, rear leg height, and rib and bone) in Xinjiang Brown cattle. Note: WW is wither width; HLHC is half of leg circumstance; RLH is rear leg height; and RAB is rib and bone. Different colors represent different chromosomes, and the number on the horizontal coordinate is the chromosome number.

**Figure 7 ijms-25-10557-f007:**
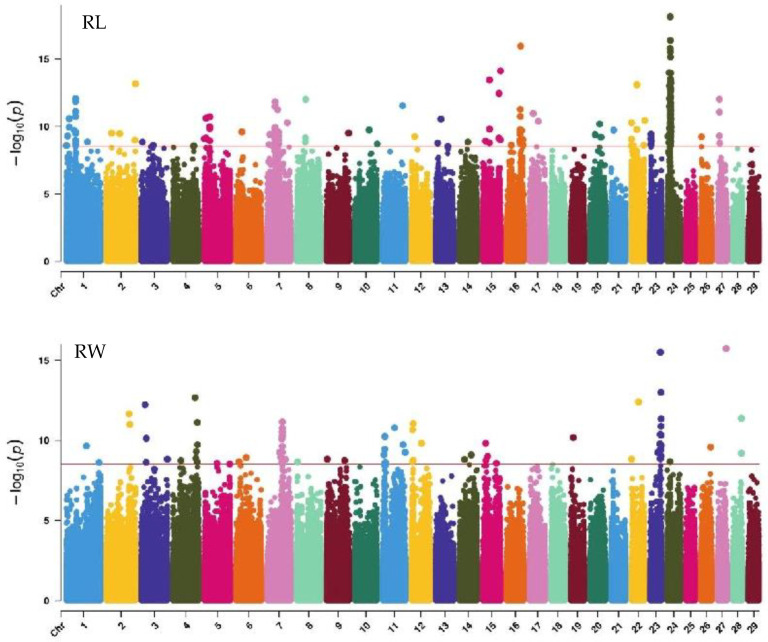
Manhattan plots of rump traits (rump length, rump width, and rump angle) in Xinjiang Brown cattle. Note: RL is rump length; RW is rump width; and RA is rump angle. Different colors represent different chromosomes, and the number on the horizontal coordinate is the chromosome number.

**Figure 8 ijms-25-10557-f008:**
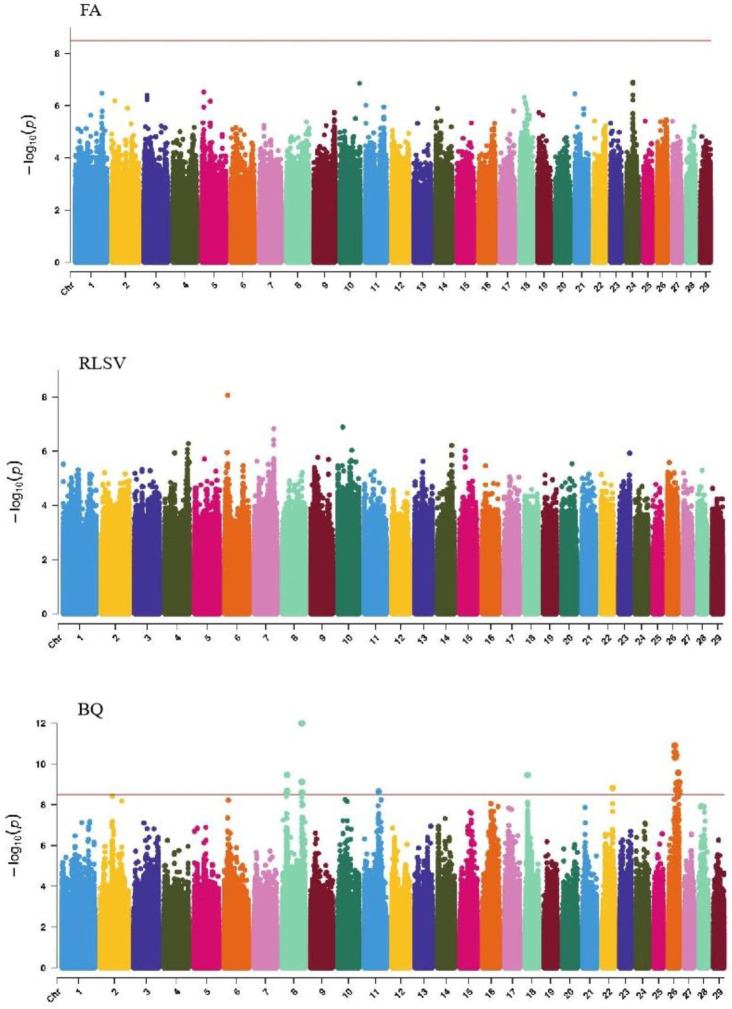
Manhattan plots of rump traits (feet angle, rear leg side view, bone quality, and rear leg rear view) in Xinjiang Brown cattle. Note: FA is feet angle; RLSV is rear leg side view; BQ is bone quality; and RLRV is rear leg rear view. Different colors represent different chromosomes, and the number on the horizontal coordinate is the chromosome number.

**Figure 9 ijms-25-10557-f009:**
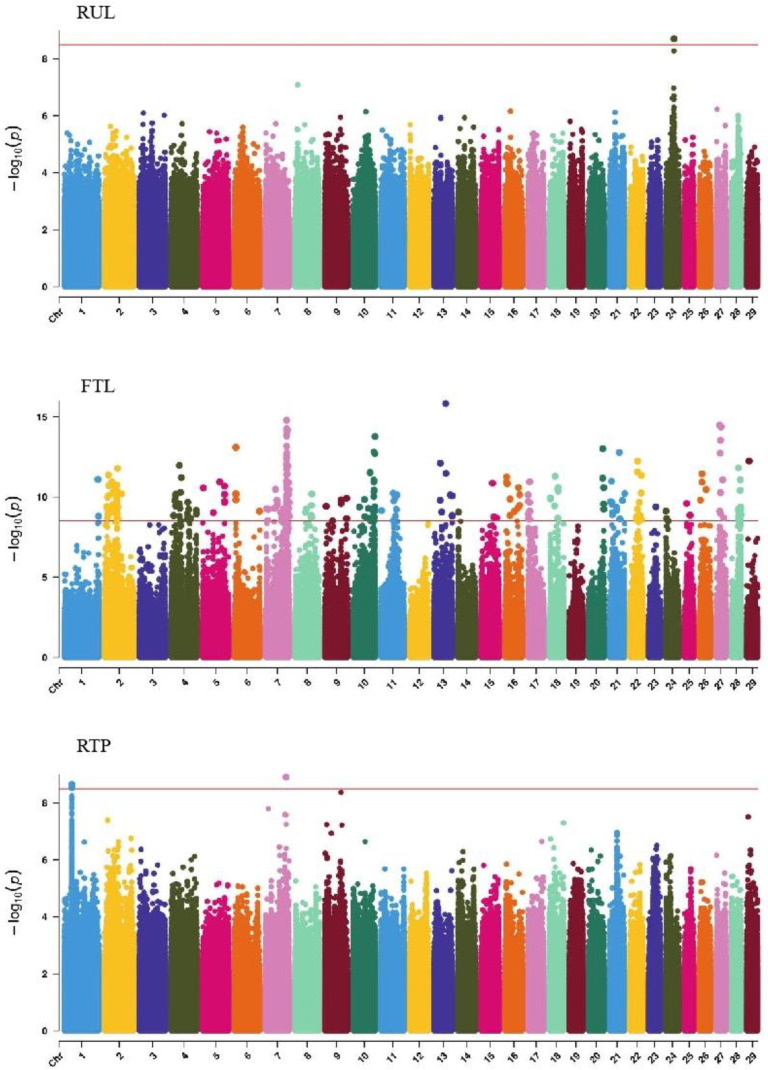
Manhattan plots of udder traits (rear udder length, fore teat length, and rear teat placement) in Xinjiang Brown cattle. Note: RUL is rear udder length; FTL is fore teat length; and RTP is rear teat placement. Different colors represent different chromosomes, and the number on the horizontal coordinate is the chromosome number.

**Figure 10 ijms-25-10557-f010:**
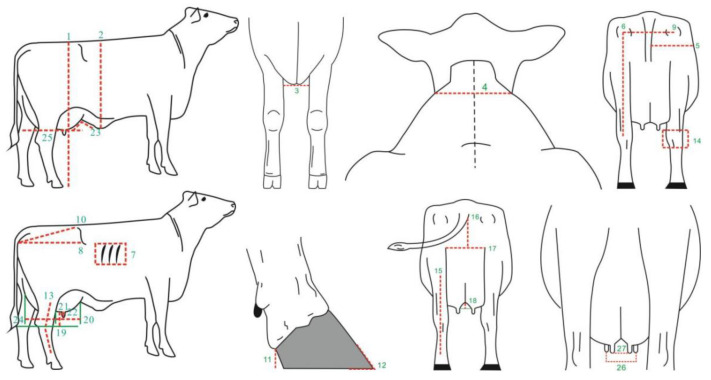
Description and measurement sites of 27 body conformation traits. Note: 1. stature (cm); 2. body depth (cm); 3. chest width (cm); 4. withers width (cm); 5. hind leg half circumference (cm); 6. rear leg height (cm); 7. rib and bone (points); 8. rump length (cm); 9. rump width (cm); 10. rump angle (cm); 11. heel depth (cm); 12. foot angle (points); 13. rear legs side view (points); 14. bone quality (points); 15. rear legs rear view (points); 16. rear udder height (cm); 17. rear udder width (cm); 18. median suspensory (cm); 19. udder depth (cm); 20. fore udder length (cm); 21. front teat length (cm); 22. front teat diameter (cm); 23. fore udder attachment (points); 24. rear udder length (points); 25. udder balance (points); 26. fore teat placement (points); and 27. rear teat placement (points).

**Table 1 ijms-25-10557-t001:** Descriptive statistics, results of variance components and heritability of body conformation traits in Xinjiang Brown cattle.

Trait	N	AVG	Std	$\sigma_{a}^{2}$	$\sigma_{p}^{2}$	$h^{2}$ ± SE
Stature (cm)	1185	139.47	6.54	20.05	13.62	0.59 ± 0.07
Body depth (cm)	1185	76.89	6.98	16.25	22.28	0.42 ± 0.07
Chest width (cm)	1185	25.45	4.62	4.07	14.26	0.22 ± 0.07
Withers width (cm)	1185	16.81	4.22	4.21	12.84	0.25 ± 0.06
Hind leg half circumference (cm)	1185	40.52	5.06	6.13	22.69	0.27 ± 0.07
Rear leg height (cm)	1185	76.56	6.23	16.50	36.18	0.46 ± 0.07
Rib and bone (points)	1185	6.34	1.23	0.33	1.42	0.24 ± 0.07
Rump length (cm)	1185	52.06	3.81	5.86	12.89	0.45 ± 0.08
Rump width (cm)	1185	21.74	3.14	4.15	9.84	0.42 ± 0.08
Rump angle (cm)	1185	6.44	3.89	6.61	14.71	0.45 ± 0.07
Heel depth (cm)	1185	4.23	1.40	0.96	1.73	0.55 ± 0.06
Foot angle (points)	1185	5.03	1.06	0.44	1.09	0.40 ± 0.07
Rear legs side view (points)	1185	4.97	1.06	0.29	1.13	0.25 ± 0.07
Bone quality (points)	1185	6.11	0.87	0.27	0.68	0.40 ± 0.07
Rear legs rear view (points)	1185	5.48	1.11	0.29	1.21	0.24 ± 0.07
Rear udder height (cm)	1185	24.79	5.00	6.77	23.68	0.29 ± 0.08
Rear udder width (cm)	1185	11.80	2.96	3.25	6.74	0.48 ± 0.07
Median suspensory (cm)	1185	3.16	1.72	0.96	2.88	0.33 ± 0.07
Udder depth (cm)	1185	10.32	5.80	6.65	23.91	0.28 ± 0.07
Fore udder length (cm)	1185	16.80	4.49	2.92	14.50	0.20 ± 0.07
Front teat length (cm)	1185	5.32	1.60	0.83	2.48	0.33 ± 0.08
Front teat diameter (cm)	1185	2.62	0.58	0.03	0.30	0.09 ± 0.05
Fore udder attachment (points)	1185	5.31	1.53	0.37	2.25	0.16 ± 0.06
Rear udder length (points)	1185	4.68	1.47	0.23	1.74	0.13 ± 0.06
Udder balance (points)	1185	4.85	0.74	0.004	0.53	0.01 ± 0.03
Fore teat placement (points)	1185	4.26	1.05	0.27	1.07	0.25 ± 0.06
Rear teat placement (points)	1185	5.20	1.05	0.34	1.08	0.31 ± 0.07

Note: N is number of data; AVG is average value; STD is standard deviation; $\sigma_{a}^{2}$ is additive variance; $\sigma_{p}^{2}$ is phenotypic variance; $h^{2}$ is heritability; SE is standard error.

**Table 2 ijms-25-10557-t002:** Eigenvalues, proportional variance, and cumulative variance explained by factor analysis of the phenotypic values of body conformation traits in Xinjiang Brown cattle.

Factor	Xinjiang Brown Cattle		
	EV	PV	CV
F1	4.36	16.14	16.14
F2	2.68	9.91	26.04
F3	1.88	6.96	33.01
F4	1.82	6.74	39.75
F5	1.59	5.89	45.64
F6	1.38	5.12	50.76
F7	1.23	4.57	55.33
F8	1.02	3.79	59.12

Note: EV is eigenvalue; PV is proportional variance; CV is cumulative variance.

**Table 3 ijms-25-10557-t003:** Results of variance components and heritability estimates for each factor in Xinjiang Brown cattle.

Factor	Xinjiang Brown Cattle		
	$\sigma_{a}^{2}$	$\sigma_{p}^{2}$	$h^{2}$ ± SE
F1	0.39	0.83	0.47 ± 0.08
F2	0.66	0.91	0.73 ± 0.06
F3	0.23	0.73	0.32 ± 0.07
F4	0.28	0.97	0.29 ± 0.06
F5	0.24	0.95	0.25 ± 0.06
F6	0.15	0.92	0.17 ± 0.06
F7	0.44	0.92	0.48 ± 0.07
F8	0.20	0.91	0.23 ± 0.07

Note: $\sigma_{a}^{2}$ is additive variance; $\sigma_{p}^{2}$ is phenotypic variance; $h^{2}$ is heritability; SE is standard error.

**Table 4 ijms-25-10557-t004:** Descriptive summary of GWAS results for body frame traits in Xinjiang Brown cattle.

Traits	Chromosome	Leading SNP Position (bp)	Candidate Gene	Distance (bp)	*p*-Value	PVE
ST	6	37,156,064	*LAP3*	within	5.17 × 10^−13^	0.113
ST	6	37,083,116	*IBSP*; *TRNAA-CGC*	189,261; 35,190	8.96 × 10^−13^	0.111
ST	6	37,178,336	*FAM184B*	2739	1.10 × 10^−12^	0.110
ST	6	37,173,382	*MED28*	within	4.37 × 10^−12^	0.104
ST	6	37,526,812	*LCORL*	within	6.32 × 10^−11^	0.092
ST	6	37,316,093	*NCAPG*	within	1.21 × 10^−10^	0.090
CW	3	7,396,186	*NOS1AP*	within	5.85 × 10^−12^	0.091
CW	6	107,343,143	-	-	6.89 × 10^−10^	0.074

Note: ST is stature of body frame traits; CW is chest width body frame traits; and PVE is the proportion of phenotypic variation of body frame traits explained by SNP.

**Table 5 ijms-25-10557-t005:** Descriptive summary of GWAS results for muscularity traits in Xinjiang Brown cattle.

Traits	Chromosome	Leading SNP Position (bp)	Candidate Gene	Distance (bp)	*p*-Value	PVE
WW	1	55,232,433	*TRNAE-UUC*	130,076	5.08 × 10^−15^	0.116
HLHC	13	56,406,710	*CDH26*	136,548	2.26 × 10^−11^	0.086
HLHC	13	4,980,813	*BTBD3*	460,810	4.17 × 10^−10^	0.076
RLH	15	75,555,654	*SYT13*; *LOC112441655*	215,956; 110,178	5.84 × 10^−22^	0.171
RLH	1	138,331,308	*CPNE4*	within	2.89 × 10^−19^	0.150
RLH	13	31,113,024	*RSU1*	within	6.30 × 10^−19^	0.147
RLH	8	111,522,169	*LOC107133158*; *MYT1L*	41,739; 144,213	1.75 × 10^−18^	0.144
RLH	13	23,276,492	*DNAJC1*; *TRNAN-GUU*	127,664; 67,644	1.06 × 10^−16^	0.130
RLH	1	124,804,486	*C1H3orf58*	61,606	1.34 × 10^−16^	0.129
RLH	13	6,103,499	*LOC107133022*; *SPTLC3*	332,139; 495,947	7.75 × 10^−16^	0.123
RLH	15	59,958,836	-	-	2.39 × 10^−15^	0.119
RLH	23	46,221,522	*OFCC1*	within	2.37 × 10^−15^	0.119
RLH	1	119,678,994	*AGTR1*	157,693	3.97 × 10^−15^	0.117
RLH	1	123,789,868	-	-	6.31 × 10^−14^	0.107
RLH	9	65,139,247	*TBX18*; *CEP162*	449,486; 31,139	1.02 × 10^−13^	0.106
RLH	8	111,303,902	*EIPR1*	134,290	1.48 × 10^−13^	0.104
RLH	4	17,840,320	-	-	1.61 × 10^−13^	0.104
RLH	13	24,864,358	*KIAA1217*	within	3.32 × 10^−13^	0.102
RLH	11	56,062,392	*CTNNA2*	within	4.51 × 10^−13^	0.100
RLH	1	140,514,230	*DSCAM*	within	5.52 × 10^−13^	0.100
RLH	9	65,891,078	*THEMIS*	within	6.42 × 10^−13^	0.099
RLH	12	81,171,137	-	-	8.14 × 10^−13^	0.098
RLH	2	109,054,387	-	-	9.41 × 10^−13^	0.098
RLH	1	114,421,449	*LOC100299503*; *RAP2B*	46,289; 132,996	1.50 × 10^−12^	0.096
RLH	1	96,976,406	*SLC7A14*; *CLDN11*	1,663; 23,553	1.61 × 10^−12^	0.096
RLH	11	88,683,898	*ID2*; *LOC107132953*	73,732; 438,273	2.55 × 10^−12^	0.094
RLH	13	31,901,223	*ST8SIA6*; *HACD1*	5,214; 155,236	2.62 × 10^−12^	0.094
RLH	1	48,448,033	-	-	4.33 × 10^−12^	0.092
RLH	15	57,328,966	*ANO3*	within	4.32 × 10^−12^	0.092
RLH	11	56,624,752	*REG3G*; *REG3A*	15,039; 92,048	5.34 × 10^−12^	0.092
RLH	9	67,321,559	*LAMA2*	within	6.82 × 10^−12^	0.091
RLH	6	101,641,411	*PTPN13*	within	6.92 × 10^−12^	0.091
RLH	15	81,168,554	*CTNND1*; *LOC101905743*	155,927; 28,840	7.43 × 10^−12^	0.090
RLH	7	100,724,011	-	-	9.06 × 10^−12^	0.090
RLH	21	9,397,972	-	-	1.37 × 10^−11^	0.088
RLH	4	93,221,222	*AHCYL2*	within	2.26 × 10^−11^	0.086
RLH	15	80,292,897	*LOC538839*; *TRNAG-UCC*	38,975; 37,581	3.27 × 10^−11^	0.085
RLH	6	94,004,701	*PAQR3*; *NAA11*	326,894; 25,528	3.45 × 10^−11^	0.085
RLH	29	4,556,310	*LOC112444871*	93,571	4.05 × 10^−11^	0.084
RLH	20	60,499,302	-	-	4.09 × 10^−11^	0.084
RLH	10	95,382,534	-	-	4.28 × 10^−11^	0.084
RLH	12	11,810,266	*VWA8*	within	4.42 × 10^−11^	0.084
RLH	17	31,137,557	-	-	4.40 × 10^−11^	0.084
RLH	15	67,951,958	-	-	4.98 × 10^−11^	0.084
RLH	3	111,827,535	*CSMD2*	within	6.44 × 10^−11^	0.083
RLH	15	65,795,201	*SLC1A2*	within	1.98 × 10^−10^	0.078
RLH	10	4,554,455	*TMED7*; *CDO1*	83,142; 105,267	2.26 × 10^−10^	0.078
RLH	4	112,604,658	*KRBA1*; *ZNF467*	17,660; 9,721	2.66 × 10^−10^	0.077
RLH	11	51,254,398	-	-	7.67 × 10^−10^	0.074
RLH	17	36,293,976	*FSTL5*	within	9.66 × 10^−10^	0.073
RLH	4	99,573,605	*FAM180A*; *LUZP6*	27,538; 174,805	1.01 × 10^−9^	0.073
RLH	4	94,507,456	*TSGA13*; *KLF14*	39,240; 13,682	1.06 × 10^−9^	0.072
RLH	1	132,107,446	*TRNAG-CCC*; *IL20RB*	446,839; 59,937	1.40 × 10^−9^	0.071
RLH	2	111,640,369	*KCNE4*; *SCG2*	425,040; 128,519	2.42 × 10^−9^	0.069

Note: WW is wither width of muscularity traits; HLHC is half of leg circumstance of muscularity traits; RLH is rear leg height of muscularity traits; and PVE is the proportion of phenotypic variation of muscularity traits explained by SNP.

**Table 6 ijms-25-10557-t006:** Descriptive summary of GWAS results for rump traits in Xinjiang Brown cattle.

Traits	Chromosome	Leading SNP Position (bp)	Candidate Gene	Distance (bp)	*p*-Value	PVE
RL	24	14,476,249	*LOC112444152*; *TRNAC-GCA*	154,623; 310,450	7.13 × 10^−19^	0.147
RL	24	15,816,993	*TRNAK-UUU*	1,486,023	6.94 × 10^−16^	0.123
RL	22	24,594,726	*CNTN4*; *CNTN6*	314,747; 348,640	7.88 × 10^−14^	0.107
RL	24	11,685,839	*CDH7*	757,611	9.51 × 10^−14^	0.106
RL	1	43,127,665	*DCBLD2*	Within;	8.44 × 10^−13^	0.098
RL	27	9,412,286	-	-	9.53 × 10^−13^	0.098
RL	11	92,037,253	*LOC525099*	198,207	2.85 × 10^−12^	0.094
RL	7	37,062,050	*SEMA6A*	254,929	3.24 × 10^−12^	0.093
RL	16	63,035,090	*CACNA1E*; *ZNF648*	85,495; 134,485	5.31 × 10^−12^	0.092
RL	5	24,082,338	*CEP83*	within	1.90 × 10^−11^	0.087
RL	1	13,348,848	-	-	2.55 × 10^−11^	0.086
RL	23	3,921,483	*DST*; *COL21A1*	153,438; 104,699	3.74 × 10^−10^	0.076
RW	23	46,200,692	*OFCC1*	within	3.17 × 10^−16^	0.126
RW	7	70,019,926	*EBF1*	264,327	7.08 × 10^−12^	0.091

Note: RL is rump length; RW is rump width and PVE is proportion of variance in phenotype explained by SNP.

**Table 7 ijms-25-10557-t007:** Descriptive summary of GWAS results for feet and leg traits in Xinjiang Brown cattle.

Traits	Chromosome	Leading SNP Position (bp)	Candidate Gene	Distance (bp)	*p*-Value	PVE
BQ	8	94,701,622	*ABCA1*	within	1.02 × 10^−12^	0.097
BQ	26	31,905,369	*ADRA2A*	315,371	1.27 × 10^−11^	0.088
BQ	26	48,348,379	*MGMT*	470,290	2.70 × 10^−10^	0.077
BQ	26	50,656,701	*KNDC1*	within	2.05 × 10^−9^	0.070
RLRV	3	15,899,137	*KCNN3*; *ADAR*	3997; 45,742	1.12 × 10^−9^	0.072

Note: BQ is bone quality; RLRV is rear leg rear view; and PVE is proportion of variance in phenotype explained by SNP.

**Table 8 ijms-25-10557-t008:** Descriptive summary of GWAS results for udder traits in Xinjiang Brown cattle.

Traits	Chromosome	Leading SNP Position (bp)	Candidate Gene	Distance (bp)	*p*-Value	PVE
RUL	24	37,881,063	*DLGAP1*	4567	1.97 × 10^−9^	0.071
FTL	7	97,778,551	*RGMB*	228,107	1.60 × 10^−15^	0.120
FTL	7	98,389,255	*CHD1*	234,752;	1.03 × 10^−14^	0.114
FTL	6	7,951,064	*LOC526064*; *TRAM1L1*	80,971; 322,188	7.91 × 10^−14^	0.107
FTL	4	37,920,167	*PCLO*; *CACNA2D1*	77,791; 233,071	1.07 × 10^−12^	0.097
FTL	28	32,737,766	*KCNMA1*	within	1.54 × 10^−12^	0.096
FTL	2	60,780,813	*TRNAC-GCA*; *CXCR4*	65,918; 469,336	1.66 × 10^−12^	0.096
FTL	2	19,033,714	*PDE11A*	within	4.26 × 10^−12^	0.092
FTL	21	9,202,570	*ARRDC4*	453,612	1.03 × 10^−11^	0.089
FTL	7	107,467,422	*FBXL17*	within	1.26 × 10^−11^	0.088
FTL	5	100,589,449	*LOC112446744*; *OVOS2*	60,780; 90,536	2.13 × 10^−11^	0.087
FTL	2	39,424,814	*TRNAC-ACA*; *GPD2*	121,972; 222,653	5.40 × 10^−11^	0.083
FTL	4	21,149,728	*ARL4A*	382,191	6.51 × 10^−11^	0.083
FTL	9	73,967,273	*PDE7B*	within	1.49 × 10^−10^	0.080
FTL	17	964,943	*CPE*; *MIR2285J-1*	243,497; 400,878	1.89 × 10^−10^	0.079
FTL	2	14,998,986	*CERKL*	within	5.34 × 10^−10^	0.075
FTL	10	93,577,847	*LOC112448582*	200,597	7.33 × 10^−10^	0.074
FTL	13	83,115,519	*TRNAG-GCC*; *CBLN4*	173,774; 135,360	1.46 × 10^−9^	0.071
FTL	4	74,525,760	*CFAP69*; *GTPBP10*	22,464; 7241	2.41 × 10^−9^	0.069
RTP	7	95,405,216	*ELL2*; *PCSK1*	18,652; 341,087	1.25 × 10^−9^	0.072
RTP	1	34,175,889	*CADM2*	73,138	2.23 × 10^−9^	0.070

Note: RUL is rear udder length; FTL is fore teat length; RTP is rear teat placement; and PVE is proportion of variance in phenotype explained by SNP.

## Data Availability

The study did not report any data.

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
