# Peer review of "Genome-Wide Association Study on Body Conformation Traits in Xinjiang Brown Cattle"

_ijms, 2024, doi:10.3390/ijms251910557_

Round 1

Reviewer 1 Report

Comments and Suggestions for Authors

Zhang et al., report the results of GWAS, genetic parameter and factor analysis on 27 body confirmation traits in Xinjian Brown cattle, while the work is interesting, i have significant concerns.  
Some concerns are, its not clear how many animals were finally used in the GWAS. In the materials the authors report sequencing 496 animals, while in other places its reported that 1185 animals were used in the GWAS.
Clarify how many animals were genotyped ? Moreover, in the GWAS model, there are a lot of levels in the covariates, for a small dataset this seems excessive and depress statistical power.  
I couldn't find a genetic correlation figure between the 27 traits, i recommend the authors add and discuss about that. Overall my concerns are the small sample size and the large number of levels in the model. Moreover, it isnt clear
to me if the focus of the paper was to verify if factors / PCs could be used for effectively reducing data dimension, while capturing significant pehnotypic / genetic variance, then why did the GWAS focus on individual traits, rather than the factors ?

Line 52: Why is reducing the analysis of correlated traits concerning for breeders, when are traits are correlated are we not capturing the same genetic effect ?
Line 53: Reducing the dimension of data to a few variables using factor analysis or PCA makes a lot more sense when you have many uncorrelated traits ?
Line 67 - 68 :- " Change to just "The genetic evaluation models ...." , genetic evaluation is the generally the standard way of reporting.
Line 72: "The body conformation traits of Xinjiang Brown cattle include those of Holstein cattle and incorporate indicators reflecting meat performance. " What does the author
mean by Xinjian Brown cattle include those of Holstein cattle ? do you mean its a composite / cross bred cattle breed with holstein genetics ?, if so it must be reported that the cattle is a cross bred and the parental breeds reported
Line 81 :- How was LD calcualted, what measure was used ? D' or R2
Line 100 : The authors report that 496 cattle were resequenced, is the data deposited in a public repository ? You had reported in the previous section that 1,185 animals were utilized for gwas ?
Line 156:- Did you do a single maker gwas or haplotype gwas ?
Line 197:- When you had measure LD and define the region in LD as an associated Loci, what is the rationale for looking for genes in the flanking region, instead of within the LD block ?

it isnt clear if the  Were the correlation between trait measures and Factors or loadings from PCA and factors?  Please clarify this.
I recommend the authors to provide a supplementary file, showing the distribution of the traits.
I suggest that authors move the qq-plots into a grid, its very difficult to figure out to which trait each qq-plot belogs, at-least in the version is reviewed, the qq-plot title was shifted.
Figure 6: Again the title for each manhattan plot was shifted and overlapped and wasn't clear which manhattan plot was from which trait gwas. I recommend, the authors keep only significant gwa manhattan plots in the main manuscript and move the ones with no significant association to the supplementary.
Why wasn't a gwas with the 8 factors identified done ? also i wonder why the gwas results were not presented first followed by the genetic parameter,that seems more natural way of doing these kind of analysis.

Author Response

Thank you for your hard review and good suggestions. I value your opinions and suggestions very much. The following modifications have been made to the article according to your comments and suggestions:

Point 1: Some concerns are, its not clear how many animals were finally used in the GWAS. In the materials the authors report sequencing 496 animals, while in other places its reported that 1185 animals were used in the GWAS. Clarify how many animals were genotyped ?

Response 1: We agree with expert, the sentence has already changed into "This study utilized pedigree information and phenotypic ( 1,185 records ) and genomic data ( The resequencing of 496 Xinjiang Brown cattle generated approximately 74.9 billion reads. ) of Xinjiang Brown cattle to estimate the genetic parameters, perform factor analysis, and conduct a genome-wide association study (GWAS) for these traits. Our results indicated that most traits exhibit moderate to high heritability. " 

Point 2: Moreover, in the GWAS model, there are a lot of levels in the covariates, for a small dataset this seems excessive and depress statistical power.

Response 2: Your suggestion is very good. As for the impact of reducing the number of covariates on GWAS analysis, GWAS analysis with different levels of covariates from few to many will be conducted in the future to explore the differences between them.

Point 3: I couldn't find a genetic correlation figure between the 27 traits, i recommend the authors add and discuss about that. Overall my concerns are the small sample size and the large number of levels in the model. Moreover, it isnt clear.

Response 3: Relevant reports have been made.

[1]Huang Yuechuan, Zhang Hailiang, Xu Wei. et al. Estimation of genetic parameters of body size and appearance traits of dairy cows in Ningxia region[J].Chinese animal husbandry and Veterinary medicine, 2024,51(07):2908-2922. DOI:10.16431/j.cnki.1671-7236.2024.07. 017.

[2]Liu Songbai, Yi Jianming, Yan Bangfu. et al. Estimation of genetic parameters of body size traits of Holstein cattle in Wuhan area[J]. Hubei agricultural sciences, 2009, 48(07): 1690-1693+1735.

Point 4: to me if the focus of the paper was to verify if factors / PCs could be used for effectively reducing data dimension, while capturing significant pehnotypic / genetic variance, then why did the GWAS focus on individual traits, rather than the factors ?

Response 4: Since 27 body traits in the selection index of Xinjiang brown cattle have not been fully included, the factor analysis part of this paper is also to explore the differences between factors and single traits. The purpose of GWAS for single traits was to explore markers for 27 body traits for future GWAS and GS combined studies, hoping to improve the accuracy of genome selection in Xinjiang brown cattle. Because the complete genetic evaluation of 27 body types is still required to describe the bulls, GWAS analysis for 8 factors was not performed.

Point 5: Line 52: Why is reducing the analysis of correlated traits concerning for breeders, when are traits are correlated are we not capturing the same genetic effect ?

Response 5: Because in breeding selection, the highly correlated traits can be selected indirectly to the remaining highly correlated traits by selecting one of them. I have added it in the paper.

Point 6: Line 53: Reducing the dimension of data to a few variables using factor analysis or PCA makes a lot more sense when you have many uncorrelated traits ?

Response 6: Only factors with eigenvalues > 1 were considered for analysis. since the variance explained by principal components with residual eigenvalues <1 is smaller than the example, a further analysis of them is not considered.

Point 7: Line 67 - 68 :- " Change to just "The genetic evaluation models ...." , genetic evaluation is the generally the standard way of reporting.

Response 7: Change “The genomic genetic evaluation models” to “The genetic evaluation models”.

Point 8: Line 72: "The body conformation traits of Xinjiang Brown cattle include those of Holstein cattle and incorporate indicators reflecting meat performance. " What does the author mean by Xinjian Brown cattle include those of Holstein cattle ? do you mean its a composite / cross bred cattle breed with holstein genetics ?, if so it must be reported that the cattle is a cross bred and the parental breeds reported

Response 8: Change “The body conformation traits of Xinjiang Brown cattle include those of Holstein cattle and incorporate indicators reflecting meat performance.” to “The body conformation traits of evaluation method Xinjiang Brown Cattle (dual-purpose breed) is formed by absorbing the linear scoring method of Chinese Holstein cattle (dairy breed), referring to the “ code of practice of type classification in Chinese Holstein ” published in 2017, and introducing the special body size evaluation method of meat breeds.”

Point 9: Line 81 :- How was LD calcualted, what measure was used ? D' or R2

Response 9: The LD is denoted by R2 ().

Point 10: Line 100 : The authors report that 496 cattle were resequenced, is the data deposited in a public repository ? You had reported in the previous section that 1,185 animals were utilized for gwas ?

Response 10: 496 animals were utilized for GWAS. 

Point 11: Line 156:- Did you do a single maker gwas or haplotype gwas ?

Response 11: Single maker GWAS.

Point 12: Line 197:- When you had measure LD and define the region in LD as an associated Loci, what is the rationale for looking for genes in the flanking region, instead of within the LD block ?

Response 12: Thank you very much for your suggestion. In this study, we did not conduct linkage disequilibrium analysis of SNP sites, and we only annotated the candidate genes upstream and downstream of the discovered SNPS. We cannot understand what LD you are referring to and would appreciate your further response.

Point 13: it isnt clear if the Were the correlation between trait measures and Factors or loadings from PCA and factors?  Please clarify this.

Response 13: The factor is the conversion of 27 body type traits by dimensionality reduction, so that the correlation of phenotypic traits with strong correlation is weakened after dimensionality reduction. as shown in the figure.

Point 14: I recommend the authors to provide a supplementary file, showing the distribution of the traits.

Response 14: Figure 2 shows it.

Point 15I suggest that authors move the qq-plots into a grid, its very difficult to figure out to which trait each qq-plot belogs, at-least in the version is reviewed, the qq-plot title was shifted.

Response 15: Thank you for your suggestion. It has been modified according to your suggestion.

Point 16:Figure 6: Again the title for each manhattan plot was shifted and overlapped and wasn't clear which manhattan plot was from which trait gwas. I recommend, the authors keep only significant gwa manhattan plots in the main manuscript and move the ones with no significant association to the supplementary.

Response 16: Thank you for your suggestion. It has been modified according to your suggestion.

Point 17: Why wasn't a gwas with the 8 factors identified done ? also i wonder why the gwas results were not presented first followed by the genetic parameter,that seems more natural way of doing these kind of analysis.

Response 17: The purpose of GWAS for single traits was to explore markers for 27 body traits for future GWAS and GS combined studies, hoping to improve the accuracy of genome selection in Xinjiang brown cattle. Because the complete genetic evaluation of 27 body types is still required to describe the bulls, GWAS analysis for 8 factors was not performed.

Thank you for your work on this paper again. Should you have any questions regarding this revision, please feel free to contact us.

Have a lovely day!

Kind regards,

Menghua Zhang

Reviewer 2 Report

Comments and Suggestions for Authors

The article presents in a very clear and interesting way the genetic determinants of body conformation traits in Xinjiang Brown Cattle. The work layout is typical, the analytical methods are properly selected and the results are clearly presented and discussed. My minor comments mainly concern the Material and Methods section. The total number of animals should be given at the beginning of the chapter. The names traits are imprecise and misleading. What is the difference between hind leg and rear leg? And what is the difference between fore udder and front udder? What is rib and bone? It is better to use anatomical terms such as thoracic limb, pelvic limb, or forelimb, hindlimb. I hope that the authors will easily remove these minor errors and I eagerly await the publication.

Author Response

Thank you for your hard review and good suggestions. I value your opinions and suggestions very much. The following modifications have been made to the article according to your comments and suggestions:

Point 1:  The total number of animals should be given at the beginning of the chapter. 

Response 1: We agree with expert, the sentence has already changed into "This study utilized pedigree information and phenotypic ( 1,185 records ) and genomic data ( The resequencing of 496 Xinjiang Brown cattle generated approximately 74.9 billion reads. ) of Xinjiang Brown cattle to estimate the genetic parameters, perform factor analysis, and conduct a genome-wide association study (GWAS) for these traits. Our results indicated that most traits exhibit moderate to high heritability. " 

Point 2: The names traits are imprecise and misleading. What is the difference between hind leg and rear leg? And what is the difference between fore udder and front udder? What is rib and bone? It is better to use anatomical terms such as thoracic limb, pelvic limb, or forelimb, hindlimb.

Response 2: Thank you for your very good suggestion, we have added the diagram in response to your suggestion. Do you think this will make the traits clearer?

Figure 1  Description and measurement sites of 27 body conformation traits

Note: 1. Stature (cm); 2. Body depth (cm); 3. Chest width (cm); 4. Withers width (cm); 5. Hind leg half circumference (cm); 6. Rear leg height (cm); 7. Rib and bone (points); 8. Rump length (cm); 9. Rump width (cm); 10. Rump angle (cm); 11. Heel depth (cm); 12. Foot angle (points); 13. Rear legs side view (points); 14. Bone quality (points); 15. Rear legs rear view (points); 16. Rear udder height (cm); 17. Rear udder width (cm); 18. Median suspensory (cm); 19. Udder depth (cm); 20. Fore udder length (cm); 21. Front teat length (cm); 22. Front teat diameter (cm); 23. Fore udder attachment (points); 24. Rear udder length (points); 25. Udder balance (points); 26. Fore teat placement (points); 27. Rear teat placement (points).

Thank you for your work on this paper again. Should you have any questions regarding this revision, please feel free to contact us.

Have a lovely day!

Kind regards,

Menghua Zhang

Reviewer 3 Report

Comments and Suggestions for Authors

The authors identified 102 significant SNPs and detected their associations with 12 body conformation traits in Xinjiang Brown cattle.

Generally, the manuscript is good, the introductions is informative. The methods are clearly described, and the discussion is deep enough. However, I have some small comments.

The investigated Xinjiang Brown cattle breed could be introduced in more detailed. A more detailed description of the breed, relationship to other breeds and some characteristics, in the introduction would be desirable.

L 44-45: In the context of this work, the term breeding is more appropriate compared to mating

L 187: The authors choose 500kb flanking regions! Please provide supporting references.

L 238-234: The sentence must be revised. The authors identified 142, 0, and 2 significant SNPs, for which traits?

L 302: The sentence must be revised.

L 328: The sentence must be revised.

The References list must be revised according to journal requirements.

Regarding In the citations, from the total of 79 citation, 3 are self-citations by authors.

I recommend the manuscript to be considered for publication after minor revision. 

Author Response

Thank you for your hard review and good suggestions. I value your opinions and suggestions very much. The following modifications have been made to the article according to your comments and suggestions:

Point 1: The investigated Xinjiang Brown cattle breed could be introduced in more detailed. A more detailed description of the breed, relationship to other breeds and some characteristics, in the introduction would be desirable.

Response 1: I've already added "The breeding industry of Xinjiang Brown cattle accounts for a large proportion of the local economic development as well as farmers’ and herders’ income. In 2023, the population of Xinjiang Brown cattle reached 2 million. "

Point 2: L 44-45: In the context of this work, the term breeding is more appropriate compared to mating. â€¨

Response 2: Change “mating” to “breeding”.

Point 3: L 187: The authors choose 500kb flanking regions! Please provide supporting references.

Response 3: Thank you very much for your suggestion. There are many ways that SNP can affect the expression of candidate genes. SNP may be located in cis-regulatory regions such as promoter region and enhancer region of a gene, or it may regulate the expression of trans-acting factors, etc., or the gene has a starting or ending position in the upstream and downstream 500 KB of SNP. Therefore, it is necessary to expand the annotation range of candidate genes to find potentially affected candidate genes. The upstream and downstream range of 500kb is reasonable, and this screening range is widely cited in cattle related GWAS studies.

[30]Purfield, D. C., Bradley, D. G., Kearney, J. F., & Berry, D. P. Genome-wide association study for calving traits in Holstein–Friesian dairy cattle. Animal (Cambridge, England), 2014. 8(2): p. 224-235.

Point 4:L 238-234: The sentence must be revised. The authors identified 142, 0, and 2 significant SNPs, for which traits?.

Response 4: We agree with expert, the abstract has already changed into "GWAS for body conformation traits (Stature, Body depth, Chest width) in Xinjiang Brown cattle identified 142, 0, and 2 significant SNPs." 

Point 5: L 302: The sentence must be revised.

Response 5: the sentence has already changed into " Note: ST is stature of body frame traits; CW is chest width body frame traits; PVE is the proportion of phenotypic variation of body frame traits explained by SNP. "

Point 6: L 328: The sentence must be revised.

Response 6: the sentence has already changed into " Note: WW is wither width of muscularity traits; HLHC is half of leg circumstance of muscularity traits; RLH is rear leg height of muscularity traits; PVE is the proportion of phenotypic variation of muscularity traits explained by SNP. "

Point 7: The References list must be revised according to journal requirements.

Response 7: It has been modified according to the requirements of the magazine reference format.

Thank you for your work on this paper again. Should you have any questions regarding this revision, please feel free to contact us.

Have a lovely day!

Kind regards,

Menghua Zhang

Round 2

Reviewer 1 Report

Comments and Suggestions for Authors

Thank you for your response and for making the changes recommended.